# Hydrate Formation with the Memory Effect Using Classical Nucleation Theory

**I. Yucel Akkutlu \*, Emre Arslan and Faisal Irshad Khan**

Department of Petroleum Engineering, College of Engineering, Texas A&M University, College Station, TX 77843, USA; emrearslan@tamu.edu (E.A.); fikhan@tamu.edu (F.I.K.)
\* Correspondence: akkutlu@tamu.edu

**Abstract:** Methane hydrate formation is analytically studied in the presence of the water memory effect using the classical nucleation theory. The memory effect is introduced as a change in nucleation site from a three-dimensional heterogenous nucleation on a solid surface with cap-shaped hydrate clusters (3D-HEN) to a two-dimensional nucleation on the solid hydrate residue surface with monolayer disk-shaped hydrate clusters (2D-NOH). The analysis on the stationary nucleation of methane hydrate under isobaric conditions shows that the memory effect caused an average decrease of 4.4 K in metastable zone width, or subcooling. This decrease can be erased at higher dissociation temperatures ($\Delta T > 17.2$ K) due to a decrease in the concentration of 2D-NOH nucleation sites. Moreover, the probability of hydrate formation is estimated for the purpose of quantifying risk associated with methane hydrate formation in the presence of the memory effect.

**Keywords:** hydrate; memory effect; nucleation; probability





## 1. Introduction

Hydrates are categorized under the compound group named clathrates. They consist of a guest molecule of one substance surrounded by host molecules of another substance. Host molecules form a cage structure, and the guest molecule resides in these structures freely without forming bonds [1]. Natural gas hydrates, in particular those composed of one natural gas molecule such as methane or carbon dioxide enclosed in water molecules, have attracted the attention of researchers from various fields. Three main reasons exist for this interest: natural gas hydrates occur in pipes and create a flow barrier [2], natural gas hydrates are present in large volumes as a potential energy resource [2], and the hydrate clathrates could offer an alternative storage mechanism for carbon sequestration in the ocean and in geological formations [2]. The focus of these studies is to understand the mechanisms of hydrate growth and dissociation.

Hydrate dissociation is an endothermic process and heat must be supplied by the system and transferred to the location to break the hydrogen bonds between the water molecules of the clathrates and to overcome the van der Waals forces between the natural gas molecule inside the cage and the water molecules [2]. Because a majority of researchers agree on hydrate dissociation being a heat-transfer limited process [3–9], hydrate plug removal methods from pipes are mainly concentrated on delivering heat to the hydrate through thermal stimulation or chemical (thermodynamic inhibitor) injection.

A phenomenon recognized by many researchers as the water "memory effect", however, may cause natural gas hydrates to form easier after being dissociated or melted down [10–27]. In these studies, the memory effect is observed as a shorter induction time during a constant subcooling or as a smaller subcooling phase during a linear cooling ramp.

Many studies have reported an explanation of the dominant physical mechanism of the memory effect since it was first identified by Vysniauskas and Bishnoi as the effect of water history on hydrate formation [28]. Structural memory, which refers to the presence of residual structures in the liquid water after hydrate dissociation, was one of the first

theories to explain the memory effect [10,16–18,29]. Then, Rodger theorized the guest supersaturation model in which the supersaturation of the gas molecules upon dissociation of hydrates was considered as the reason for the memory effect [11]. This theory was later advanced by Bagherzadeh et al., who claimed nano-bubbles existed in the bulk liquid following the hydrates' dissociation [30]. This was later confirmed by Uchida et al. by showing the formation of micro- and nano-bubbles in the liquid after the dissociation and this was linked to the memory effect [31]. Zeng et al. postulated the impurity imprinting theory which postulated the presence of heterogeneous nucleation and hence the necessity of solid walls for realization of the memory effect [15]. Similarly, interfacial gaseous states theory states that the nano-bubbles should be forming on the solid wall as opposed to forming in the bulk liquid because heterogeneous nucleation on the walls is always easier—less energy is required—than homogeneous nucleation in the bulk liquid phase [32–34]. In a more recent study, it was reported that the memory effect depends on the thermal history of the water [26].

All these studies are considered fundamental in solving the mystery behind the memory effect during secondary hydrate formation. Based on these studies and considering the classical nucleation theory [35], it is safe to say that the hydrate memory effect is attributed to increased nucleation rates. However, due to the stochastic nature of the nucleation process, it has been difficult to make direct experimental observations. Adams et al. recently performed hydrate crystallization experiments using sand-pack [36]. In this study, we will attempt to develop a computational methodology to investigate the nucleation rates caused by the memory effect using a new modeling approach based on the classical nucleation theory, CNT. The latter is chosen for this purpose due to its widespread use in defining the nucleation of clathrate hydrates and its analytical simplicity [37]. Combining the theory with the so-called shortest path of hydrate formation method [38], a sample computation of generating a hydrate equilibrium curve with and without the memory effect will be provided. Lastly, the conditions for erasing the memory effect will also be investigated.

Primary hydrate formation involves a solid surface such as a pipe wall or a grain of rock for the crystal nucleation and growth, as seen in Figure 1a. In the case of 3D-HEN, cap-shaped hydrate clusters form at the interface between the solid surface and water with a wetting angle $\theta$. In 2D nucleation on previously formed hydrate clusters (2D-NOH), however, hydrates form easier because there is no solid–water interface and hence less of an energy barrier needs to be overcome, as seen in Figure 1b [35]. It is theorized here, that after the dissociation of hydrate, the residual 2D hydrate clusters with monolayer height can act as nucleation sites with increased nucleation rates compared to three-dimensional heterogeneous nucleation (3D-HEN).

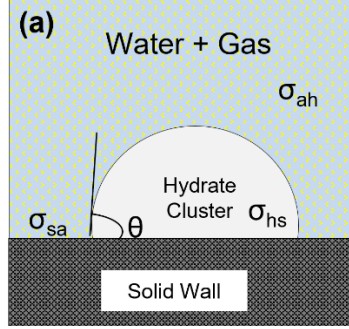 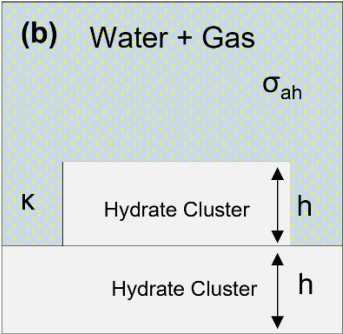

**Figure 1.** (**a**) Representation of 3D-HEN and (**b**) representation of 2D-NOH.

## 2. Computational Method

The methodology used in this study involves (1) generating the hydrate equilibrium curve of methane using PVTSim®, a software that can be used to construct the curve; (2) calculating nucleation rates at different temperatures using Classical Nucleation Theory; (3) calculating the growth rates at different temperatures; (4) calculating induction times

at different temperatures; (5) generating the spinodal curve and metastable zone width (MSZW); and (6) generating the probability map of having "hydrate growth" using the shortest path to hydrate formation method [38].

### 2.1. Generation of the Hydrate Equilibrium Curve

PVTsim® Nova from Calsep Inc in Houston, TX, USA, was used to generate the methane hydrate equilibrium curve. (SRK Peneloux model and a composition of 95.2 mol% water and 4.7 mol% methane was chosen for illustrative purposes.) The resulting curve shown in Figure 2 was fitted with a third-degree polynomial function:

$$P_e = 0.003607 \times T_e^3 - 3.004 \times T_e^2 + 834.3 \times T_e - 7.727 \times 10^4, \tag{1}$$

where $P_e$ is equilibrium pressure in MPa at $T_e$, the equilibrium temperature in Kelvin. This hydrate equilibrium curve can be validated with experiments with multiple data points (pressure and temperature). However, due to the lack of an experimental setup, the output from the PVTsim is used in this study.

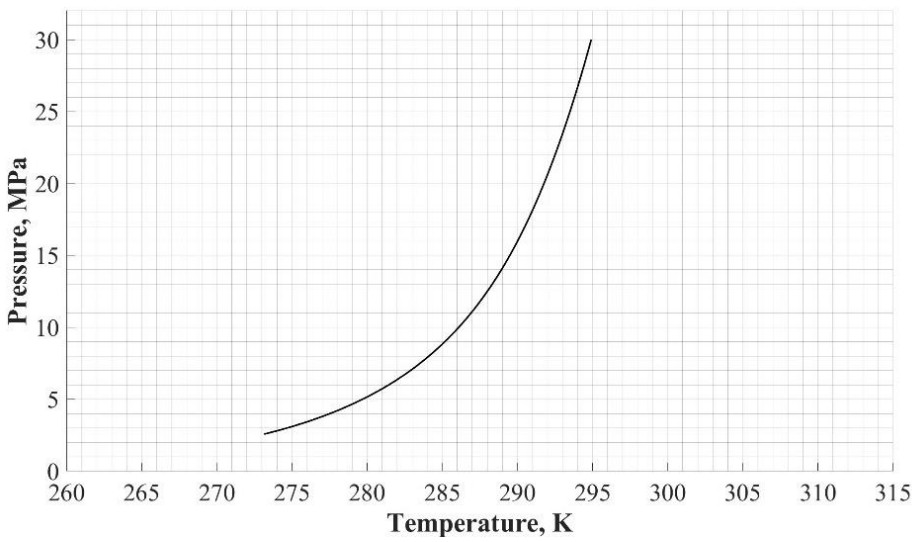

**Figure 2.** Methane hydrate equilibrium curve.

### 2.2. Calculation of Nucleation Rates

In a continuously stirred tank environment, when the methane–water system is cooled down at a constant pressure to a temperature below $T_e$, 3D-HEN applies. In this first cooling cycle, there is no memory effect and the cap-shaped hydrate clusters form across three interfaces: solution/hydrate, solution–solid, and hydrate/solid interfaces (Figure 3a) [35]. Therefore, the effective specific surface energy is calculated using the following equations:

$$\sigma_{ef} = \psi \sigma_{ah}, \tag{2}$$

$$\psi = \left[ (1/4)(2 + cos\theta)(1 - cos\theta)^2 \right]^{1/3}, \tag{3}$$

$$cos\theta = (\sigma_{sa} - \sigma_{hs})/\sigma_{ah}. \tag{4}$$

In Equations (2)–(4), $\sigma_{ef}$ is the effective specific surface energy, $\psi$ is a factor between 0 and 1, $\theta$ is the wetting angle between 0 and 180 degrees, $\sigma_{sa}$ is the specific surface energy for the solution/solid interface, $\sigma_{hs}$ is the specific surface energy for the hydrate/solid interface, and $\sigma_{ah}$ is the specific surface energy for the solution/hydrate interface. $\sigma_{ah}$ is assumed, approximately, as 20 mJ/m$^2$, which is the specific surface energy of an ice/water interface, and it was assumed to remain approximately constant at any given hydrate formation pressure and temperature [39].

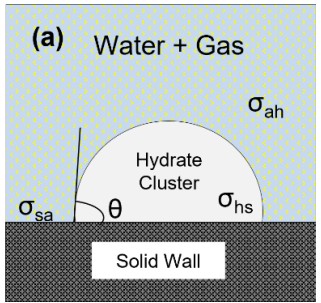 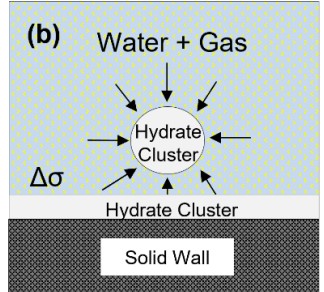 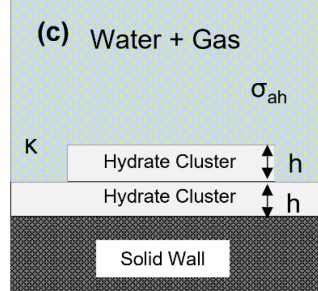

**Figure 3.** (**a**) Hydrate formation in the first cooling cycle (no memory effect), (**b**) hydrate dissociation, and (**c**) hydrate formation in the second cooling cycle (memory effect).

The nucleation rate (in $m^{-3}\ s^{-1}$) of 3D-HEN for the stationary nucleation of one component phase under isobaric conditions is given as [35]

$$J_{3D-HEN} = A\,\exp(\Delta s_e \Delta T/kT)\exp\left(-4c^3 v_h^2 \sigma_{ef}^3/27k\Delta s_e^2 T\Delta T^2\right), \tag{5}$$

where $A$ is kinetic factor, $\Delta s_e$ is the entropy of hydrate dissociation at $T_e$, $T$ is the system temperature in Kelvin, $k$ is the Boltzmann constant, $c$ is shape factor, $v_h$ is the volume of a spherical hydrate building unit, $a_h\ (=(9\pi v_h^2/16)^{1/3})$ is the cross-sectional area of a spherical-shaped hydrate building unit, and $d_h\ (=(6v_h/\pi)^{1/3})$ is the diameter of a spherical-shaped hydrate building unit. $\Delta s_e = \Delta h_e/T_e$, where $\Delta h_e$ is the enthalpy of dissociation of methane and the values found in the literature are used [40,41]. The kinetic factor ($A$) is assumed to be independent of driving force because the change in the attachment frequency of building units to the nucleus is negligible within the pressure and temperature range of hydrate formation [39]. $\Delta T = T_e - T$ is defined as the temperature difference between the equilibrium temperature and the system temperature. If $T_e > T$, then $\Delta T$ is called subcooling. If $T_e < T$, then $\Delta T$ is called superheating. The corresponding values of these parameters are given in Table 1.

**Table 1.** Parameters used in the nucleation rate calculations.

| Parameter | Unit | Value | Reference |
|---|---|---|---|
| $A$ | $m^{-3}\ s^{-1}$ | $4\times10^{26}$ | [39] |
| $k$ | $m^2\ g\ s^{-2}\ K^{-1}$ | $1.38\times10^{-23}$ | [39] |
| $\Delta s_e$ | J/K | $3.07\times10^{-22}$ | [39] |
| $c$ | - | $(36\pi)^{1/3}$ | [39] |
| $C_0$ | $m^{-2}$ | $10^{19}$ | [35] |
| $p_e$ | Mpa | 2.6 | [39] |
| $\sigma_{ah}$ | $mJ/m^2$ | 20 | [39] |
| $v_h$ | $nm^3$ | 0.216 | [39] |
| $\alpha_h$ | $nm^2$ | 0.440 | [39] |
| $d_h$ | nm | 0.744 | [39] |
| $v_0$ | $nm^3$ | 0.216 | [39] |
| $\alpha_0$ | $nm^2$ | 0.330 | [35] |
| $d_0$ | nm | 0.650 | [35] |
| $h$ | nm | 0.650 | [35] |

Hydrates dissociate radially, and therefore the heat is transferred through the melted water to the inner portions of the hydrate [2]. This type of dissociation results in a cooling region near the solid walls during the melting of the hydrate [42]. After the first cooling cycle, if the system is heated up under isobaric conditions to melt down the hydrate, a negative driving force ($\Delta\mu$) can be effective near the solid walls due to this cooling effect. When $\Delta\mu < 0$ and $\Delta\sigma < 0$, hydrate structures can deposit on the solid surfaces via 2D undersaturation nucleation (2D-UNS) [35], creating nucleation sites for 2D nucleation

on the hydrate (2D-NOH) (Figure 3b). The nucleation rate of 2D-UNS is given by the following equations:

$$J_{2D-UNS} = A' \, exp(\Delta s_e \Delta T / kT) exp[-B/(\Delta s_e \Delta T - a_0 \Delta \sigma)], \tag{6}$$

$$A' = A''/\eta(T), \tag{7}$$

$$A'' = \left[ b(kT)^{1/2} (\Delta s_e \Delta T)^{1/2} / 6\pi d_0 v_0 \right] C_0 (p_e v_0 / kT), \tag{8}$$

$$\eta(T) = 4.47 \times 10^{-7} exp[1234.6/(T - 122.3)], \tag{9}$$

$$B = b^2 \kappa^2 / 4kT, \tag{10}$$

$$b = 2(\pi a_0)^{1/2}, \tag{11}$$

$$\kappa = \sigma_{ah} h, \tag{12}$$

where $A'$ is the kinetic factor, $\Delta \sigma$ is the specific surface energy, $\kappa$ is the specific edge energy (equivalent of specific surface energy), $\eta(T)$ is the temperature dependence of the viscosity of water, $v_0$ is the volume of a disk-shaped hydrate building unit, $a_0 (= v_0/d_0)$ is the cross-sectional area of a disk-shaped hydrate building unit, $d_0 \left( = (4v_0/\pi)^{1/3} \right)$ is the diameter of a disk-shaped hydrate building unit, $h = d_0$ is the height of the disk-shaped hydrate building unit, $C_0$ is the concentration of nucleation sites, and $p_e$ is the phase-equilibrium pressure of the liquid.

The 2D undersaturation nucleation (2D-UNS) could be at the heart of explaining the memory effect, because with increasing superheating ($\Delta T$), the nucleation rate of 2D-UNS decreases (Figure 4). Assuming a linear relationship between $J_{2D-UNS}$ and $C_0$, concentration of nucleation sites ($C_0$) also decreases with increasing $\Delta T$ (Figures 4 and 5), which can explain why the memory effect is erased at higher superheating values. The effect on MSZW is further discussed in Section 2.5.

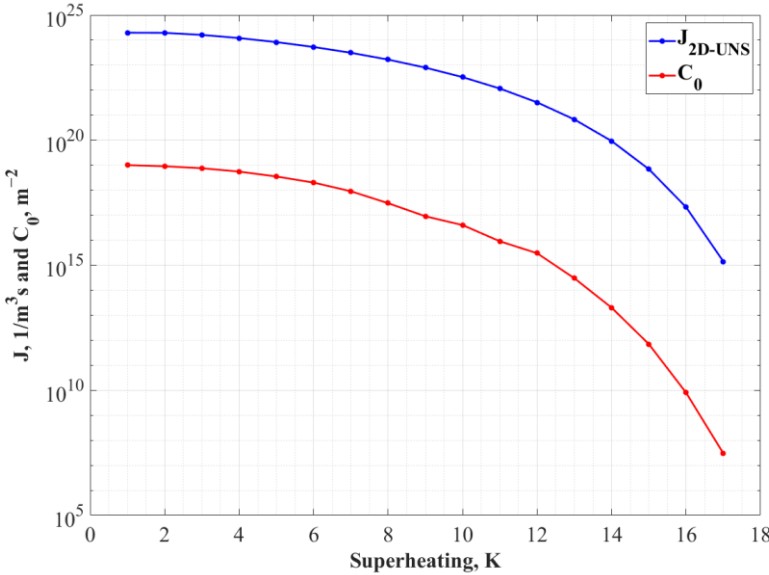

**Figure 4.** Decrease in nucleation rate of 2D-UNS and concentration of nucleation sites ($C_0$) with superheating during dissociation at $T_e$ = 293.2 K and $P_e$ = 19.4 MPa for $\Delta \sigma$ = −20 mJ/m$^2$.

In the second cooling cycle, we consider that 2D nucleation on the hydrate (2D-NOH) dominates the nucleation process, as seen in Figure 3c. The term 2D-NOH is derived specifically for the hydrate case, representing 2D nucleation on the substrate (a subtype of heterogeneous nucleation) and is based on the Classical Nucleation Theory explained by Kashchiev [35]. In this scenario, the hydrate cluster plays the role of the solid surface but there is no interface between solid and water. The creation of the lateral phase boundary

is the only energy barrier for nucleation to take place. The nucleation rate (in $m^{-3} s^{-1}$) taking place between 2D monolayer hydrate disks is given with Equation 13 [35]. The corresponding values of these parameters are given in Table 1.

$$J_{2D-NOH} = A' \, exp(\Delta s_e \Delta T / kT) exp[-B/(\Delta s_e \Delta T)], \tag{13}$$

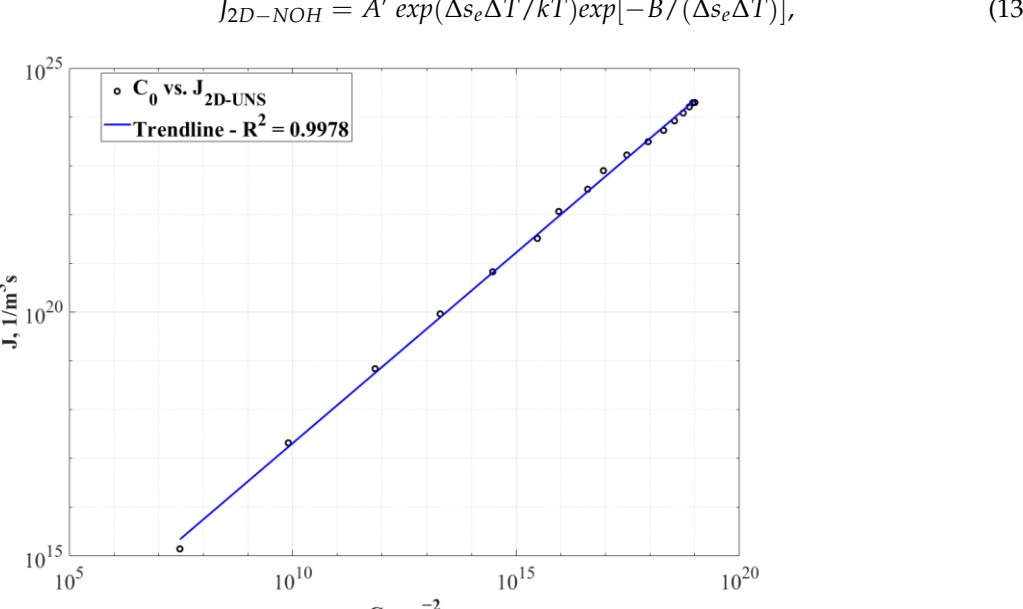

**Figure 5.** Linear relationship between $J_{2D\text{-UNS}}$ and $C_0$ at $T_e$ = 293.2 K and $P_e$ = 19.4 MPa for $\Delta\sigma = -20 \, mJ/m^2$.

As a sample calculation, using the parameters given in Table 1 at equilibrium conditions of $T_e = 293.2$ K and $P_e = 19.4$ MPa, the nucleation rates were calculated for 2D-NOH, 3D-HEN at $\theta = 60°$, and 3D-HEN at $\theta = 90°$ at different subcooling temperatures under isobaric conditions. In the extreme case of 2D-NOH, the contact angle, $\theta$, does not exist. Therefore, $\theta = 60°$ and $\theta = 90°$ were chosen in our calculations to represent 3D-HEN in order to compare against 2D-NOH. $\sigma_{ef}$ is taken as 0.0108 $J/m^2$ for $\theta = 60°$ and 0.0159 $J/m^2$ for $\theta = 90°$, respectively [39]. The results are shown on a semi-log plot in Figure 6.

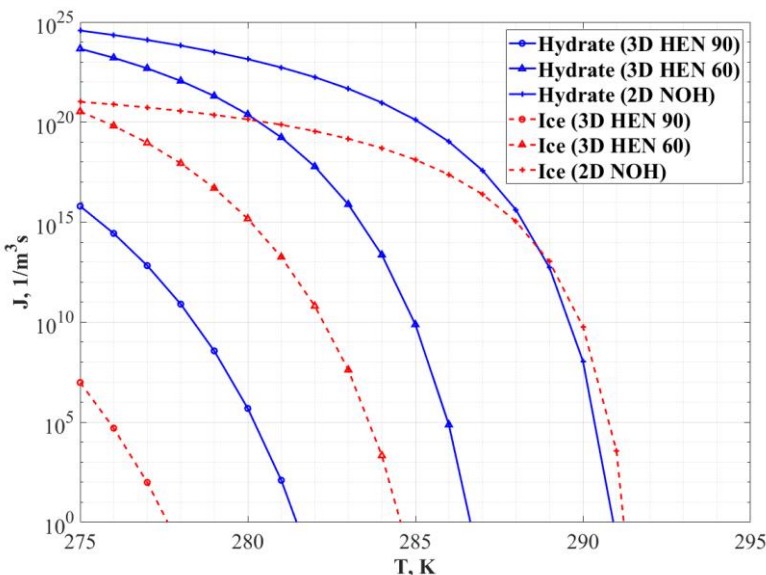

**Figure 6.** Comparison of hydrate and ice nucleation rates at $T_e$ = 293.2 K and $P_e$ = 19.4 MPa.

The memory effect is observed in the formation and melting cycles of ice as well [43]. The ice case could be a reference point for comparisons. Therefore, in Figure 6, the nucleation rate of ice formation is also included. For the calculation of these nucleation rates of ice formation, the same procedure is applied with only a few changes in the parameters used; $\Delta s_e$ (=3.07 × 10$^{-22}$ J/K) is the entropy of ice melting [35], $v_h$ (=0.033 nm$^3$) is the volume of a spherical ice building unit [35], $a_h$ (=0.124 nm$^2$) is the cross-sectional area of a spherical-shaped ice building unit [39], $d_h$ (=0.398 nm) is the diameter of a spherical-shaped ice building unit [39], $v_0$ (=0.033 nm$^3$) is the volume of a disk-shaped ice building unit [39], $a_0$ (=0.095 nm$^2$) is the cross-sectional area of a disk-shaped ice building unit [39], $d_0$ (=0.348 nm) is the diameter of a disk-shaped ice building unit [39], and $p_e$ (=0.61 kPa) is the phase-equilibrium pressure of the water [35]. The figure shows that the nucleation rates are significantly different at low temperatures.

### 2.3. Calculation of Hydrate Growth Rates

There are two types of hydrate growth mechanisms that take place for 3D-HEN (no memory effect) and 2D-NOH (memory effect). After 3D-HEN, hydrate building units are transferred across the crystallite/solution interface. The crystal surface is rough during this process, and it is called continuous growth [35]. The equation for continuous growth is given by the following equations:

$$G = \varepsilon a_h D_{ef} C_e \left( e^{\Delta\mu/kT} - 1 \right), \tag{14}$$

$$\Delta\mu = \Delta s_e \Delta T, \tag{15}$$

where $G$ is growth rate in m/s, $\varepsilon$ is the sticking coefficient of building units to the crystallite surface, $D_{ef}$ is the effective diffusion coefficient, $C_e$ is the concentration of dissolved gas at phase equilibrium, and $\Delta\mu$ is the driving force defined for isobaric conditions.

For the 2D-NOH case, on the other hand, hydrate growth can only proceed after the crystal face roughens by nucleation of 2D clusters [35]:

$$G = \varepsilon a_0 D_{ef} C_e e^{\Delta\mu/3kT} \left( e^{\Delta\mu/kT} - 1 \right)^{2/3} e^{-B/3\Delta\mu}, \tag{16}$$

This growth type is called nucleation-mediated growth, which is behind the memory effect, along with the 2D-NOH. The corresponding values of the parameters used in Equations (14)–(16) are given in Table 2. As a sample calculation, using the parameters given in Table 2 at equilibrium conditions of $T_e$ = 293.2 K and $P_e$ = 19.4 MPa, the growth rates were calculated for 2D-NOH and 3D-HEN at different subcooling temperatures under isobaric conditions. The results are shown on a semi-log plot in Figure 7. As expected, the growth rate of continuous growth is higher than the growth rate of nucleation-mediated growth [35].

**Table 2.** Parameters used in the growth rate calculations.

| Parameter | Unit | Value | Reference |
|:---:|:---:|:---:|:---:|
| $\varepsilon$ | - | 1 | [44] |
| $D_{ef}$ | μm$^2$/s | 1000 | [44] |
| $C_e$ | nm$^{-3}$ | 0.03 | [44] |
| $\alpha_h$ | nm$^2$ | 0.440 | [44] |
| $h$ | nm | 0.650 | [35] |
| $k$ | m$^2$ g s$^{-2}$ K$^{-1}$ | 1.38 × 10$^{-23}$ | [39] |
| $\Delta s_e$ | J/K | 3.07 × 10$^{-22}$ | [39] |
| $\sigma_{ah}$ | mJ/m$^2$ | 20 | [39] |
| $\alpha_0$ | nm$^2$ | 0.330 | [35] |

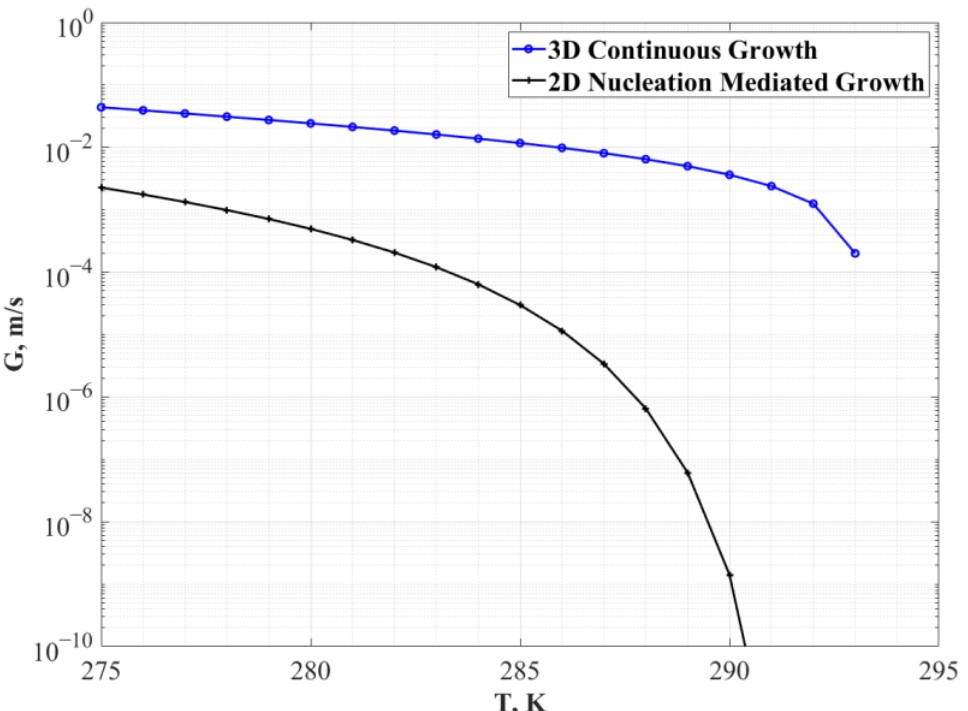

**Figure 7.** Comparison of growth rates at $T_e$ = 293.2 K and $P_e$ = 19.4 MPa.

### 2.4. Calculation of Induction Times

To quantify the effect of the memory effect on hydrate formation, induction time and/or metastable zone width values are required to be compared with and without the memory effect. Induction time can be identified as the time required for the hydrate clusters to become visible. It is a measure of the ability of a supersaturated system to stay in the state of metastability [44]. Metastability is defined as the ability of a nonequilibrium state to persist for a long period of time [2]. MSZW represents metastable one width and is used to define the metastable region in terms of subcooling under isobaric conditions. The spinodal curve marks the end of the metastable region on the left side of hydrate PT curve.

Progressive nucleation occurs when the hydrate crystallites are constantly nucleated [35]. Induction time is then calculated for progressive nucleation with the following equation:

$$t_i = ((1+3m)\alpha_d / c'G^{3m}J)^{1/(1+3m)}, \tag{17}$$

where $t_i$ is the induction time in seconds, $m$ is a factor, $\alpha_d$ is the fraction of hydrate crystallized, $c'$ is a shape factor and it is equal to $(4\pi/3)\psi$ for cap-shaped clusters [44] and 8 for disk-shaped monolayer clusters [45], $G$ is the growth rate, and $J$ is the nucleation rate. The values used for the calculation of induction times are given in Table 3.

**Table 3.** Parameters used in the induction time calculations.

| Parameter | Unit | Value |
|:---:|:---:|:---:|
| $\alpha_d$ | - | 0.01 |
| $\Psi(60°)$ | - | 0.538 |
| $c'$ (3D-60°) | - | 2.25 |
| $c'$ (2D) | - | 8 |
| $m$ | - | 1 |
| $\Psi(90°)$ | - | 0.794 |
| $c'$ (3D-90°) | - | 3.33 |

*2.5. Generating the Spinodal Curve (MSZW)*

The next step to quantify the effect of the memory effect on hydrate formation is to obtain the metastable zone width (MSZW) by generating the spinodal curve position on the left side of the hydrate equilibrium curve and measuring the distance between the two curves at constant pressure. This is accomplished by calculating the critical nucleation potential [46]. As the system is cooled down beyond the equilibrium conditions, it resists staying in the metastable region until the critical nucleation potential is reached. The nucleation potentials are given by the following equations:

$$N_{3D-HEN} = 4c^3 v_h^2 \sigma_{ef}^3, \tag{18}$$

$$N_{2D-NOH} = b^2 \kappa^2. \tag{19}$$

The instantaneous nucleation potential is calculated by dividing the critical nucleation potential by the instantaneous induction time. The difference between the equilibrium temperature and the temperature where the cumulative instantaneous nucleation potential reaches the critical nucleation potential is taken as the MSZW.

$$N_{ins,n} = \frac{N}{t_{i,n}}, \tag{20}$$

$$\Delta T = MSZW \; at \; \sum N_{ins,n} = N. \tag{21}$$

Table 4 illustrates a sample calculation for 3D-HEN with $\theta = 60°$ at $T_e = 293.2$ K and $P_e = 19.4$ MPa. Note that $N_{3D-HEN} = 2.66 \times 10^{-59}$ J·m is reached when $\Delta T = 6.9$ K.

**Table 4.** Sample calculation of the MSZW for 3D-HEN with at $T_e = 293.2$ K and $P_e = 19.4$ MPa.

| T, K | ΔT, K | J, m$^{-3}$ s$^{-1}$ | G, m/s | $t_i$, s | $N_{ins,n}$ | Cumulative $N_{ins,n}$ |
|------|-------|------|--------|----------|-------------|------------------------|
| 291 | 2.2 | $2.58 \times 10^{-205}$ | $2.39 \times 10^{-3}$ | $4.74 \times 10^{52}$ | $5.60 \times 10^{-112}$ | $5.60 \times 10^{-112}$ |
| 290 | 3.2 | $1.06 \times 10^{-83}$ | $3.62 \times 10^{-3}$ | $1.37 \times 10^{22}$ | $1.94 \times 10^{-81}$ | $1.94 \times 10^{-81}$ |
| 289 | 4.2 | $7.08 \times 10^{-38}$ | $4.97 \times 10^{-3}$ | $3.78 \times 10^{10}$ | $7.03 \times 10^{-70}$ | $7.03 \times 10^{-70}$ |
| 288 | 5.2 | $8.92 \times 10^{-16}$ | $6.44 \times 10^{-3}$ | $9.29 \times 10^{4}$ | $2.86 \times 10^{-64}$ | $2.86 \times 10^{-64}$ |
| 287 | 6.2 | $1.93 \times 10^{-3}$ | $8.04 \times 10^{-3}$ | $6.49 \times 10^{1}$ | $4.10 \times 10^{-61}$ | $4.10 \times 10^{-61}$ |
| 286 | 7.2 | $7.51 \times 10^{4}$ | $9.78 \times 10^{-3}$ | $7.09 \times 10^{-1}$ | $3.75 \times 10^{-59}$ | $3.79 \times 10^{-59}$ |
| 285 | 8.2 | $7.63 \times 10^{9}$ | $1.17 \times 10^{-2}$ | $3.48 \times 10^{-2}$ | $7.65 \times 10^{-58}$ | $8.02 \times 10^{-58}$ |
| 284 | 9.2 | $2.31 \times 10^{13}$ | $1.37 \times 10^{-2}$ | $4.15 \times 10^{-3}$ | $6.41 \times 10^{-57}$ | $7.21 \times 10^{-57}$ |
| 283 | 10.2 | $7.71 \times 10^{15}$ | $1.60 \times 10^{-2}$ | $8.66 \times 10^{-4}$ | $3.07 \times 10^{-56}$ | $3.79 \times 10^{-56}$ |
| 282 | 11.2 | $5.99 \times 10^{17}$ | $1.85 \times 10^{-2}$ | $2.62 \times 10^{-4}$ | $1.02 \times 10^{-55}$ | $1.39 \times 10^{-55}$ |
| 281 | 12.2 | $1.71 \times 10^{19}$ | $2.12 \times 10^{-2}$ | $1.02 \times 10^{-4}$ | $2.60 \times 10^{-55}$ | $3.99 \times 10^{-55}$ |
| 280 | 13.2 | $2.40 \times 10^{20}$ | $2.41 \times 10^{-2}$ | $4.79 \times 10^{-5}$ | $5.55 \times 10^{-55}$ | $9.55 \times 10^{-55}$ |
| 279 | 14.2 | $2.01 \times 10^{21}$ | $2.74 \times 10^{-2}$ | $2.56 \times 10^{-5}$ | $1.04 \times 10^{-54}$ | $1.99 \times 10^{-54}$ |
| 278 | 15.2 | $1.14 \times 10^{22}$ | $3.09 \times 10^{-2}$ | $1.51 \times 10^{-5}$ | $1.76 \times 10^{-54}$ | $3.75 \times 10^{-54}$ |
| 277 | 16.2 | $4.85 \times 10^{22}$ | $3.48 \times 10^{-2}$ | $9.66 \times 10^{-6}$ | $2.75 \times 10^{-54}$ | $6.50 \times 10^{-54}$ |
| 276 | 17.2 | $1.64 \times 10^{23}$ | $3.90 \times 10^{-2}$ | $6.54 \times 10^{-6}$ | $4.07 \times 10^{-54}$ | $1.06 \times 10^{-53}$ |
| 275 | 18.2 | $4.62 \times 10^{23}$ | $4.37 \times 10^{-2}$ | $4.63 \times 10^{-6}$ | $5.74 \times 10^{-54}$ | $1.63 \times 10^{-53}$ |

This calculation procedure was repeated at different equilibrium conditions and the results are shown in Figures 8 and 9. Since the nucleation process is a stochastic process, it is not possible to estimate the wetting angle during 3D-HEN. Therefore, the results are only representative, and it is seen that, in the case of 2D-NOH, where we claim the presence of the memory effect, the mean MSZW is smaller than the 3D-HEN. A future experimental study could validate the findings in Figures 8 and 9 by observing the changes in the induction times for the onset of hydrate formation. However, it should be noted that the induction time measurements are subject to problems due to the stochastic nature of the MSZW [47].

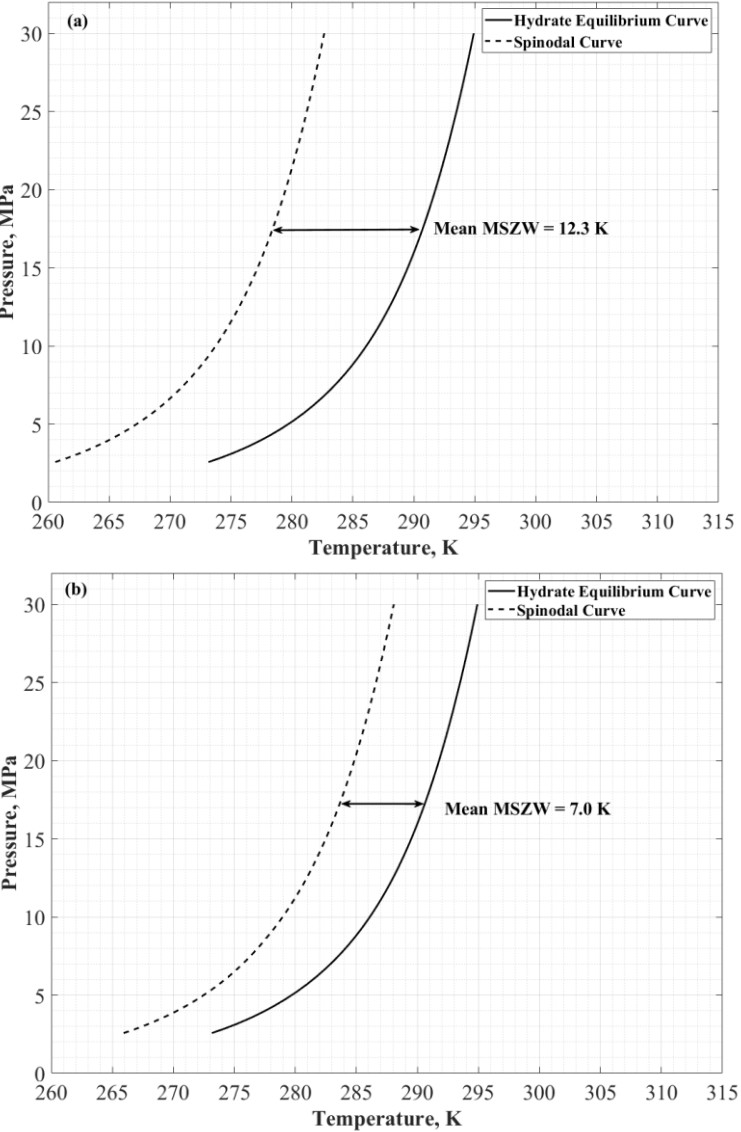

**Figure 8.** MSZW values without the memory effect (3D-HEN) for (**a**) $\theta = 90°$ and (**b**) $\theta = 60°$.

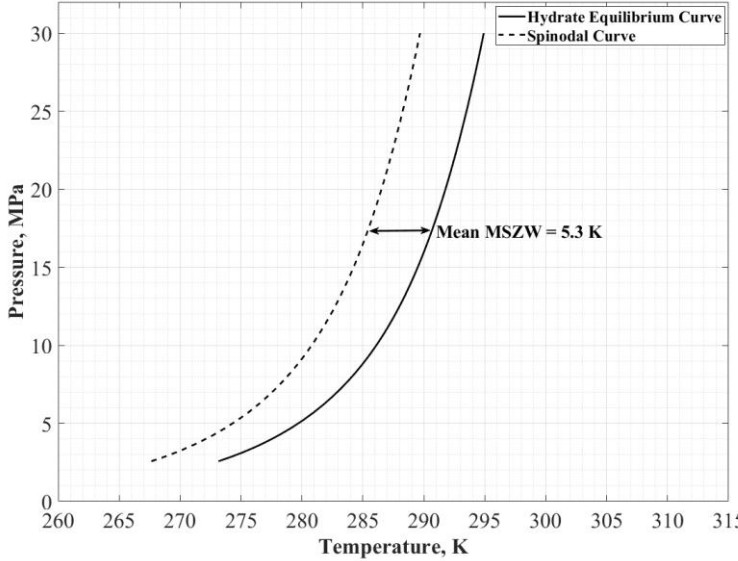

**Figure 9.** MSZW values with the memory effect (2D-NOH).

An important aspect of the memory effect in hydrates is that, after being superheated to a certain temperature, the memory effect is erased. As explained in Section 2.2, 2D undersaturation nucleation (2D-UNS) might be the reason for this. Using Equation (6), if we assume a driving force equal to the absolute value of a negative driving force ($|-\Delta\mu| = \Delta\mu$) during dissociation at $T_e = 293.2$ K and $P_e = 19.4$ MPa and assume the values as $\Delta\sigma = -20$ mJ/m$^2$ and $C_0 = 10^{19}$ m$^{-2}$, it is observed that nucleation rate values decrease with increased superheating (Figure 4). According to Equation (13) (2D-NOH), the nucleation rate is linearly proportional to the concentration of nucleation sites when the other parameters are kept constant. Taking $C_0 = 10^{19}$ m$^{-2}$ as the base value at 1 K superheating, we can extrapolate the values for the concentration of nucleation sites using the 2D-UNS nucleation rate values at $T_e = 293.2$ K and $P_e = 19.4$ MPa (Figure 5). Repeating the calculation procedure for 2D-NOH at these different $C_0$ values and at different superheating values for $\Delta\sigma = -20$ mJ/m$^2$, it is seen that the shift in the MSZW caused by the memory effect is decreased with increasing superheating values, and at 17.2 K the memory effect is erased (Figure 10).

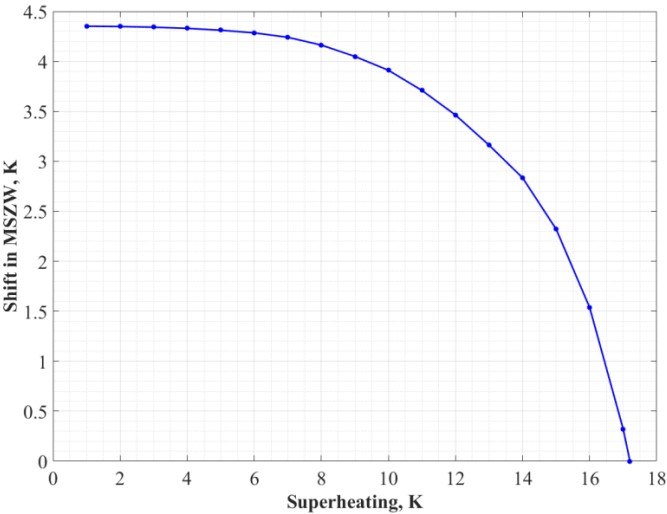

**Figure 10.** Decrease in the shift of the MSZW with increasing superheating at $T_e$ = 293.2 K and $P_e$ = 19.4 MPa.

### 2.6. Generating Probability of Hydrate Formation

In the final step, we calculated the probability of the hydrate formation on a macro level at different operating conditions. The shortest path of hydrate formation method was used to calculate different probability values on the right side of spinodal curve [37]. There are four steps in the methodology: (1) generating the hydrate equilibrium curve (and spinodal curve), (2) defining the reference temperature and pressure values, (3) using the shortest path of hydrate formation, and (4) calculating the hydrate formation probability.

The shortest path of hydrate formation method involves taking n different linear pathways from the operating conditions to the spinodal curve between two tangent lines on the spinodal curve (Figure 11). $T_{min}$ (minimum temperature) and $P_{max}$ (maximum pressure) are taken as 274.8 K and 17.2 MPa, respectively, which mark the upper and lower limits for the tangent lines. Then, the probability values are calculated according to the following formula:

$$Pr = \frac{1}{n}\sum_{i=1}^{n}\left(\frac{(T_{OP} - T_{SP})}{(T_i - T_{SP})}\right) * \left(\frac{(P_{OP} - P_{SP})}{(P_i - P_{SP})}\right), \tag{22}$$

$$Pr_i = \begin{cases} \frac{(T_{OP}-T_{SP})}{(T_i-T_{SP})}; P_i \leq P_{OP} \\ \left(\frac{(T_{OP}-T_{SP})}{(T_i-T_{SP})}\right) * \left(\frac{(P_{OP}-P_{SP})}{(P_i-P_{SP})}\right); P_i > P_{OP}, T_i \leq T_{OP}, \\ \frac{(P_{OP}-P_{SP})}{(P_i-P_{SP})}; T_i > T_{OP} \end{cases} \tag{23}$$

where $T_{OP}$ is operating temperature, $T_{SP}$ is the safe temperature, $T_i$ is the temperature on the spinodal curve, $P_{OP}$ is the operating pressure, $P_{SP}$ is the safe pressure, and $P_i$ is the pressure on the spinodal curve. Here, safe temperature and pressure represent the operating conditions with no hydrate formation probability. They are calculated using mean temperature and pressure ($T_{md}$ and $P_{md}$), which are the mean distance between the safe point and the points on the spinodal curve. Accordingly, $T_{SP}$ and $P_{SP}$ are calculated as 314.4 K and 1.7 MPa, respectively.

$$\frac{\sum_{i=1}^{n}(T_{SP} - T_i)}{n} = T_{md}, \tag{24}$$

$$\frac{\sum_{i=1}^{n}(P_{SP} - P_i)}{n} = P_{md}. \tag{25}$$

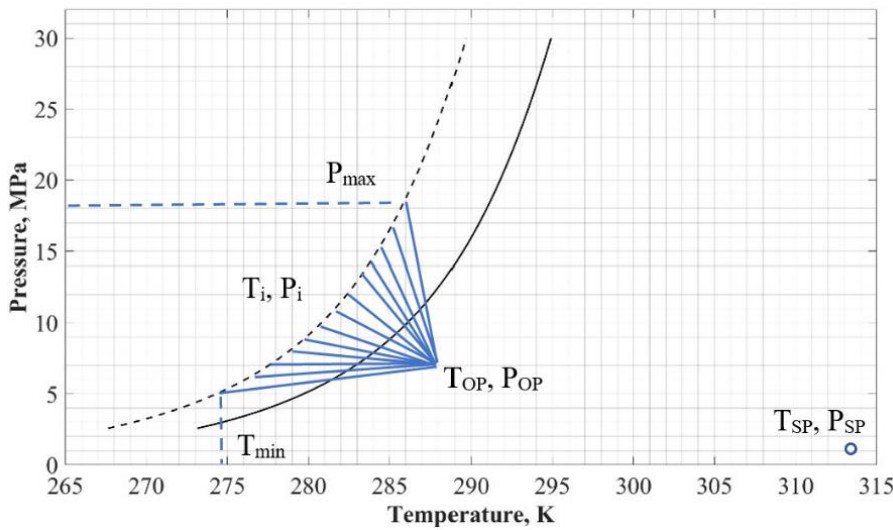

**Figure 11.** Shortest path of hydrate formation method.

MatLab was used to perform these calculations and the results are shown in Figures 12 and 13. The average growth rate on the spinodal curve is $2.22 \times 10^{-2}$ m/s for Figure 12a, $9.70 \times 10^{-3}$ m/s for Figure 12b, and $6.78 \times 10^{-7}$ m/s for Figure 13.

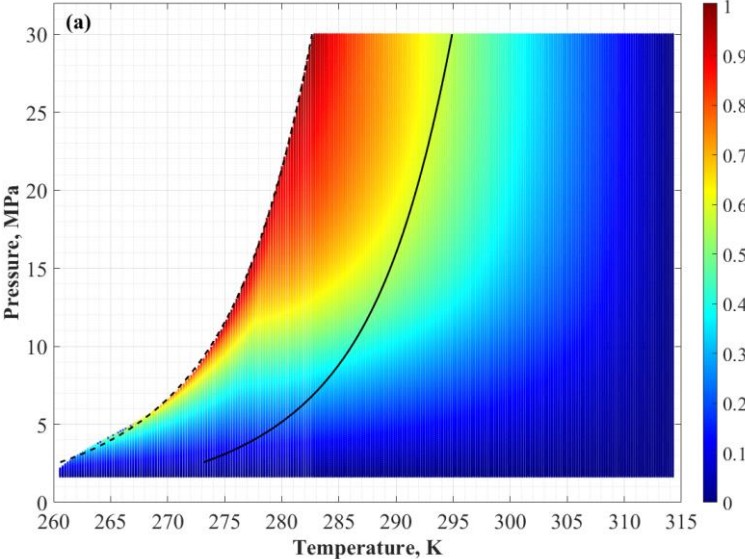

**Figure 12.** *Cont.*

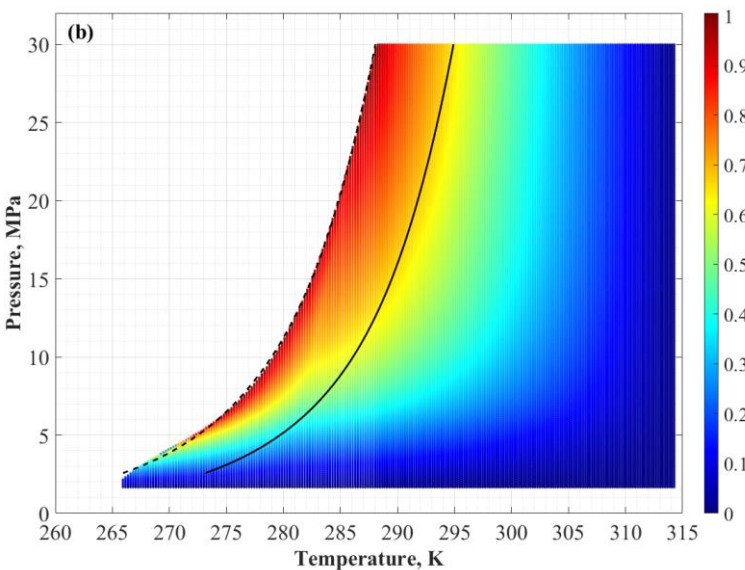

**Figure 12.** Probability values without the memory effect (3D-HEN) for (**a**) $\theta = 90°$ and (**b**) $\theta = 60°$.

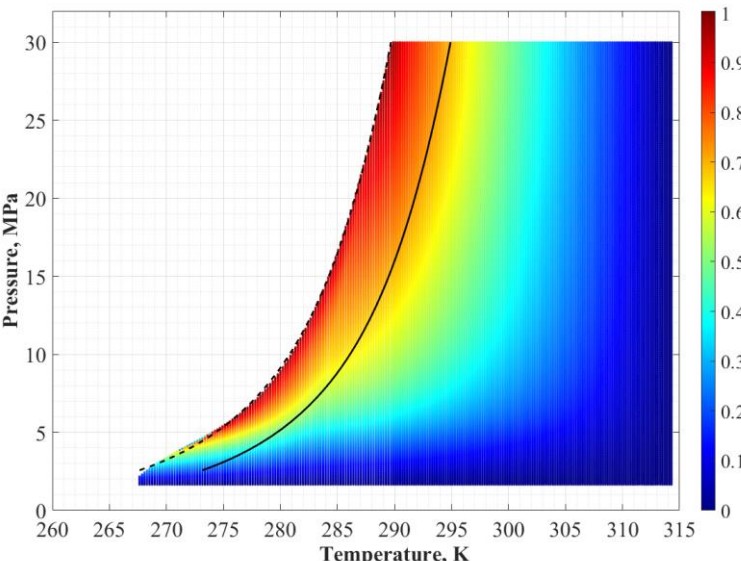

**Figure 13.** Probability values with the memory effect (2D-NOH).

## 3. Results and Discussion

Our results indicate that the change in the nucleation mechanism due to the memory effect can be explained with Classical Nucleation Theory. Accordingly, the dominant nucleation mechanism of 3D-HEN could be replaced with 2D-NOH in a second cycle of hydrate formation after the dissociation of the hydrate. This resulted in a decrease in the MSZW (subcooling) from 12.3 K ($\theta = 90°$ 3D-HEN) and 7.0 K ($\theta = 60°$ 3D-HEN) to 5.3 K (2D-NOH). It should be noted here that nucleation is a stochastic process and the two wetting angles chosen for 3D-HEN are used only as an example. It is difficult to determine how many of the nucleation events are taking place at which wetting angle. Therefore, we used these results only to quantify our theory for explaining the possible change in the nucleation mechanism which could promote the hydrate memory effect.

### 3.1. The Shift in MSZW Values

Considering the stochastic nature of nucleation events, it is difficult to make direct comparisons between the results presented here and the experimental observations made in the literature. Therefore, the methodology presented in the previous sections are only

representative. Moreover, in our calculations, methane is selected to make the math simpler with its known parameters such as surface energy, entropy, etc., whereas the experimental studies found in the literature involved the use of a mixture gas.

While keeping these challenges in mind, the explanation of the hydrate memory effect using Classical Nucleation Theory is still consistent with the literature. Taking the mean of MSZW values for 3D-HEN, a reduction of 4.4 K in the MSZW is calculated for methane hydrate because of the memory effect. This value is in parallel with the results found in the literature. An experimental study conducted with cyclopentane hydrate revealed that the onset temperature could be increased up to 4 °C in a water sample with hydrate history compared to a freshwater sample in which the onset temperature for hydrate formation is 0 °C [18]. May et al. measured the performances of different kinetic hydrate inhibitors using multi-component synthetic natural gas [48]. Their experiments showed that the subcooling values were shifted 4 K in a pure water system due to the memory effect and this shift was as high as 14 K in the worst performing inhibitor. By observing several thousand nucleation events, Sowa and Maeda concluded that a reduction of at least 4 K in subcooling was present due to the memory effect [19]. This reduction was increased with decreasing superheating temperature. In their experiments, under the single thermal stimulation dissociation pattern (STSP), Cheng et al. reported a 3.1 °C decrease in the $\Delta T$ value of methane hydrate during reformation cycles due to the memory effect [49].

### 3.2. Erasing of the Memory Effect with Increasing Superheating

It is shown here that 2D undersaturation nucleation upon hydrate dissociation can create nucleation sites for 2D-NOH which may be responsible for the hydrate memory effect. Using this concept, the sensitivity analysis conducted on the parameters of Equation (6) may have significant implications for the memory effect (Figure 14). First, $C_0$ (initial concentration of nucleation sites) and $p_e$ (the phase-equilibrium pressure of the liquid) do not play a role in the erasing of the memory effect. The changes in these parameters only change the nucleation rate of 2D undersaturation nucleation. The size and therefore the surface area ($a_0$) of the hydrate building unit is one of the important parameters controlling the erasing of the memory effect. The parameter relating to the surface energies of the solid, water, and hydrate ($\Delta\sigma$) strongly affect the superheating value at which the memory effect is erased, along with the specific surface energy term of the interface between water and the hydrate ($\sigma_{ah}$).

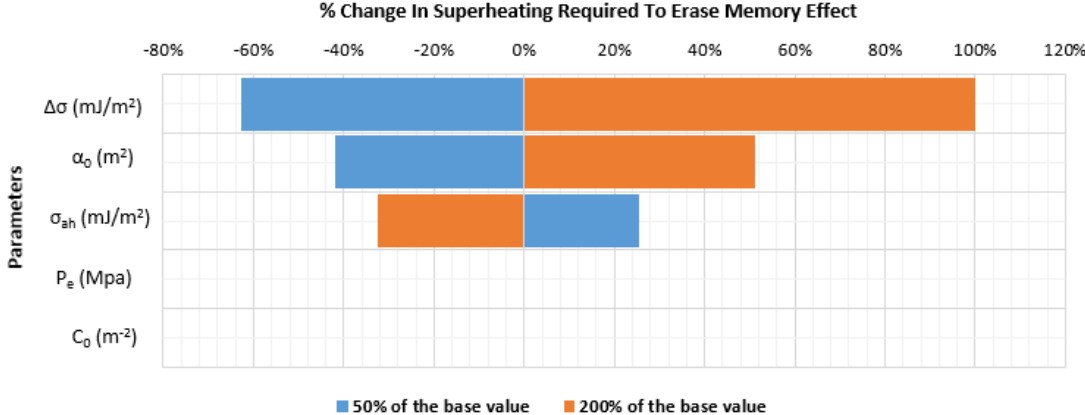

**Figure 14.** Sensitivity analysis of the parameters in Equation (6) at $T_e$ = 293.2 K and $P_e$ = 19.4 MPa.

Accordingly, the superheating temperature at which the memory effect is erased is linearly related to the specific surface energy of the solid surface. This might explain the different dissociation temperatures reported in the literature. Sowa and Maeda conducted hydrate formation experiments inside a 'glass' cell [19]. They reported that the memory effect was erased at a temperature of 311 K (37.85 °C). Wu and Zhang conducted similar research using a 'stainless steel' cell and found that the memory effect was erased at

25 °C [16]. In a series of experiments conducted inside a 'sand pack' to show the memory effect in confined spaces, Adams et al. observed that the memory effect disappeared at a dissociation temperature of 46.4 °C [36]. Table 5 summarizes these results by comparing the specific surface energies of the solids used in these studies.

**Table 5.** Comparison of dissociation temperatures at which the memory effect is erased, reported in the literature with different solid surfaces.

| Material | $\sigma$ (mJ/m$^2$) | Reference |
|---|---|---|
| Stainless Steel | 30.54 | [50] |
| Glass | 58.95 | [50] |
| Quartz | 96.00 | [51] |
| **Experimental Setup Used** | **Dissociation Temperature at which the memory effect is erased (°C)** | **Reference** |
| Stainless Steel Cell | 25.00 | [16] |
| Glass Cell | 37.85 | [19] |
| Sand pack | 46.40 | [36] |

It should be noted here that, to obtain the exact relationship between these parameters and the erasing of the memory effect, the rate of cooling (hence the amount of negative driving force) caused by the endothermic dissociation of hydrates should be known. Here, we simply assumed a 1:1 ratio, i.e., one unit of heating resulted in one unit of cooling. Although surface energies of different solids change the temperature at which the memory effect is erased, future work involving experimental studies is needed to provide empirical proof of the phenomena.

### 3.3. Probabilistic Estimation of Hydrate Formation under the Memory Effect

A probabilistic estimation for hydrate formation is also provided by using the shortest path of hydrate formation method. This way, the hydrate probability color map for methane hydrate obtained by Herath et al. is expanded to the spinodal curve [38]. Both hydrate nucleation and growth take place inside the metastable region; however, the dominant mechanism is nucleation. Once the spinodal curve is reached, spontaneous hydrate growth occurs, and hydrates become visible to the naked eye. Therefore, the knowledge of the location of this curve is important to decide if the operating conditions are safe for production without having hydrate blockage inside pipelines. The results show increased probability values between the spinodal curve and hydrate equilibrium curve, namely the metastable region, due to the memory effect.

### 4. Conclusions

Classical Nucleation Theory is used to explain a possible mechanism for the memory effect during hydrate formation/dissociation cycles. Conclusions obtained from this study are as follows:

1. Figure 3 illustrates the steps leading to the memory effect during hydrate formation and dissociation cycles;

   a. Hydrate formation occurs under isobaric conditions via 3D heterogeneous nucleation on the solid surface;

   b. Hydrate dissociates endothermically under isobaric conditions. Endothermic melting results in a cooling region near the solid surface resulting in 2D undersaturation nucleation, creating nucleation sites for the second hydrate formation cycle;

   c. During the second hydrate formation cycle, nucleation sites created on the solid surface provide the conditions for 2D nucleation on the hydrate with faster nucleation rates than 3D heterogeneous nucleation. This results in lower induction times and lower subcooling temperatures for spontaneous hydrate growth to take place, namely the memory effect;

2. Increased nucleation rates can be attributed to the change in the nucleation mechanism from 3D-HEN to 2D-NOH. Results show an average decrease of 4.4 K in subcooling temperature due to the memory effect;

3. The 2D undersaturation nucleation rate equation is used to explain the erasing of the memory effect. It has strong implications for the role of solid surfaces on which hydrate nucleation takes place. Accordingly, a solid surface with a lower specific surface energy could lead to erasing of the memory effect at lower superheating temperatures under isobaric conditions;

4. A significant increase in the hydrate formation probability is observed due to the decrease in the MSZW caused by the memory effect;

5. Future work involving experimental and/or simulation studies could support the findings presented here.

**Author Contributions:** Conceptualization, I.Y.A. and E.A.; methodology, E.A., I.Y.A. and F.I.K.; software, E.A.; validation, E.A.; formal analysis, E.A. and I.Y.A.; investigation, E.A.; resources, I.Y.A. and F.I.K.; data curation, E.A.; writing—original draft preparation, I.Y.A. and E.A.; writing—review and editing, I.Y.A. and F.I.K.; visualization, E.A.; supervision, I.Y.A. All authors have read and agreed to the published version of the manuscript.

**Funding:** This research was funded by the Mary Kay O'Connor Process Safety Center at Texas A&M University.

**Data Availability Statement:** Dataset available on request from the authors.

**Conflicts of Interest:** The authors declare no conflict of interest.

## Nomenclature

| | |
|---|---|
| $a_0$ | Cross sectional area of a disk–shaped hydrate building unit, $m^2$ |
| $a_h$ | Cross sectional area of a spherical–shaped hydrate building unit, $m^2$ |
| $\alpha_d$ | Fraction of hydrate cyrstallized |
| $\Delta\mu = \Delta s_e \Delta T$ | Driving force, J |
| $\Delta\sigma$ | Specific surface energy, $J/m^2$ |
| $\Delta h_e$ | Enthalpy of dissociation of methane |
| $\Delta s_e$ | Entropy of hydrate dissociation, J/K |
| $\Delta T = T_e - T$ | Undercooling if $T_e > T$, superheating if $T_e < T$, K |
| $\varepsilon$ | Sticking coefficient of building units to the crystallite surface |
| $\eta(T)$ | Temperature dependence of the viscosity of water |
| $\theta$ | Wetting angle between $0°$ and $180°$ |
| $\kappa$ | Specific edge energy(equivalent to $\Delta\sigma$ for 2D nucleation), J/m |
| $v_0$ | Volume of a disk–shaped hydrate building unit, $m^3$ |
| $v_h$ | Volume of a spherical–shaped hydrate building unit, $m^3$ |
| $\sigma_{ah}$ | Specific surface energy for solution–hydrate interface, $J/m^2$ |
| $\sigma_{ef}$ | Effective specific surface energy, $J/m^2$ |
| $\sigma_{hs}$ | Specific surface energy for hydrate–solid interface, $J/m^2$ |
| $\sigma_{sa}$ | Specific surface energy for solution–solid interface, $J/m^2$ |
| $\psi$ | A factor between 0 and 1 |
| $A$ | Kinetic factor for 3D nucleation, $m^{-3} s^{-1}$ |
| $A'$ | Kinetic factor for 2D nucleation, $m^{-3} s^{-1}$ |
| $C_0$ | Concentration of nucleation sites, $m^{-2}$ |
| $C_e$ | Concentration of dissolved gas at phase equilibrium, $m^{-3}$ |
| $c$ | Shape factor |
| $c'$ | Shape factor |
| $D_{ef}$ | Effective diffusion coefficient, $m^2/s$ |
| $d_0$ | Diameter of a disk–shaped hydrate building unit, m |
| $d_h$ | Diameter of a spherica–shaped hydrate building unit, m |

| | |
|---|---|
| $G$ | Growth rate, m/s |
| $h = d_0$ | Height of the disk–shaped hydrate building unit, m |
| $J$ | Nucleation rate, $\text{m}^{-3}\,\text{s}^{-1}$ |
| $J_{2D-UNS}$ | Nucleation rate of 2D undersaturation nucleation, $\text{m}^{-3}\,\text{s}^{-1}$ |
| $J_{2D-NOH}$ | Nucleation rate of 2D nucleation on own hydrate, $\text{m}^{-3}\,\text{s}^{-1}$ |
| $J_{3D-HEN}$ | Nucleation rate of 3D heterogeneous nucleation, $\text{m}^{-3}\,\text{s}^{-1}$ |
| $k$ | Boltzmann constant, $\text{m}^2\,\text{g}\,\text{s}^{-2}\,\text{K}^{-1}$ |
| $N_{2D-NOH}$ | Nucleation potential of $2D-NOH$ |
| $N_{3D-HEN}$ | Nucleation potential of $3D-HEN$ |
| $N_{ins,n}$ | Instantaneous nucleation potential |
| $P_e$ | Equilibrium pressure, MPa |
| $P_i$ | Pressure on spinodal curve, MPa |
| $P_{OP}$ | Operatingp ressure, MPa |
| $P_{SP}$ | Safe pressure, MPa |
| $p_e$ | Phase–equilibrium pressure of the liquid, MPa |
| $T$ | System temperature, K |
| $T_e$ | Equilibrium temperature, K |
| $T_i$ | Temperature on spinodal curve, K |
| $T_{OP}$ | Operating temperature, K |
| $T_{SP}$ | Safe temperature, K |
| $t_i$ | Induction time, s |

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
