# Peer review of "Hydrate Formation with the Memory Effect Using Classical Nucleation Theory"

_crystals, doi:10.3390/cryst14030243_

Round 1

Reviewer 1 Report

Comments and Suggestions for Authors

This article can be published after address my concerns as follows:

1.      Can the Hydrate equilibrium curve be validated by experiments?

2.      Table 1, where the parameters come from? How the authors obtain the kinetic factor and surface energy?

3.      Can the kinetic factor and surface energy be affected by different crystallization conditions? For example, tempearture of driving force?

4.      Table 2, where the parameters come from? Exactly right? Extracted it in any other reference?

5.      Fig.8~10, can the calculating value of MSZW be validated by experiments?

Author Response

This article can be published after address my concerns as follows:

We would like to thank Reviewer 1 for the comments

  1. Can the hydrate equilibrium curve be validated by experiments?

Yes, the hydrate equilibrium curve presented in Figure 2 can be validated by experiments. Since we did not have access to an experimental setup at the time of writing this article, we chose to use the output from PVTSim. We addressed this in Section 2.1 in the manuscript.

  1. Table 1, where the parameters come from? How the authors obtain the kinetic factor and surface energy?

We added the references for the parameters as the last column on the right in Table 1. Thank you for raising this point.

  1. Can the kinetic factor and surface energy be affected by different crystallization conditions? For example, temperature of driving force?

These two parameters are considered constant within the pressure and temperature ranges of hydrate formation. This is addressed on page 4 in the manuscript.

  1. Table 2, where the parameters come from? Exactly right? Extracted it in any other reference?

We added the references for the parameters used in Table 2.

  1. 8~10, can the calculating value of MSZW be validated by experiments?

Validating the MSZW values in Figures 8 and 9 by experiments could be possible by measuring the induction times between some pressure and temperature value on the hydrate equilibrium curve until the first visible hydrate crystals emerge. However, because the induction times is a stochastic measure within the concept of classical nucleation theory, it would not be possible to validate in a deterministic fashion. We addressed this on pages 9 and 10 in the manuscript.

Reviewer 2 Report

Comments and Suggestions for Authors

Typos in the text of the article need to be corrected. For example, in the caption to Figure 4, C_0 is indicated instead of C0; in the list of references, number 40 indicates the journal incorrectly.

Author Response

Reviewer 2

Comments and Suggestions for Authors

Typos in the text of the article need to be corrected. For example, in the caption to Figure 4, C_0 is indicated instead of C0; in the list of references, number 40 indicates the journal incorrectly.

We would like to thank Reviewer 2 for the comments.

The typos mentioned are fixed.

Reviewer 3 Report

Comments and Suggestions for Authors

The authors have presented a clear account of their model of "memory effects" for the rate of formation of methane hydrate from solutions from which methane hydrate had previously been existent. The model is based on classical nucleation theory. 

The assumptions of the model are well-defined and sufficient characterization of the model parameters and output are provided. The predictions of the model are physically meaningful and are consistent with observed experimental behaviours. 

I have a few minor comments which the authors should consider addressing:

1) Since the authors likely have all the required parameters, can they conduct a summary characterization of the memory effects for ice formation from water. This is of interest in its own right and can provide a reference for comparison with the methane hydrate formation case.

2) There are a couple of references which touch on the memory effect for hydrate formation and these should be considered for citation:

Ripmeester & Alavi, Some current challenges in clathrate hydrate science: Nucleation, decomposition and the memory effect, Curr. Opin. Solid State  Mater. Sci. 20(6), 2016, p 344-351.

Englezos et al. Kinetics of Clathrate Hydrate Processes, Chap. 11 of Clathrate Hydrates: Molecular Science and Characterization, Eds. Ripmeester & Alavi, Wiley, 2022. 
